# Reimagining Adult Religious Education and Faith Development in a Detraditionalised Ireland

**Bernadette Sweetman** 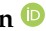

Mater Dei Centre for Catholic Education, Dublin City University, D09 DY00 Dublin, Ireland; bernadette.sweetman@dcu.ie

**Abstract:** The culture of provision of adult religious education and faith development, whereby talks or courses are made available at parish level and/or in formal educational settings, has undoubtedly dominated the Irish scene for many years. The low level of uptake of such opportunities or long-term engagement, however, coupled with the recognised decrease in regular church attendance would suggest that this culture of provision does not meet the needs of the adult population. This mismatch was a key driving force behind the inception of the Adult Religious Education and Faith Development (AREFD) project. Cognisant of cultural and societal changes, a core aim of the project was to assess this traditional culture of provision within a detraditionalised context. The present study is based on data gathered in phase two of the AREFD project consisting of fourteen semi-structured interviews and focus groups conducted between December 2019 and April 2021. The participants were involved for a number of years in adult religious education and faith development in both the Republic of Ireland and Northern Ireland and across a variety of settings. The purpose of these interviews was to gather together the rich insights from the wealth of experience of the interviewees on the practicalities and possibilities central to adult religious education. The findings affirm dissatisfaction amongst participants with the current state of AREFD in Ireland, but indicate that there is hope for the future. Fresh and innovative engagement with adults is called for. This paper outlines key themes emerging from the data which contribute to the conversation of how innovative engagement with adults can revitalise church culture in Ireland.

**Keywords:** adult religious education; adult faith development; Catholic; Republic of Ireland

## 1. Introduction

In contributing to this special volume on 'Catholic Education in Detraditionalized Cultural Contexts', it is necessary to clarify the particular aspects of this wide-ranging theme to which the present study refers. The paper arises from an ongoing research project taking place at the Mater Dei Centre for Catholic Education (MDCCE), Dublin City University (DCU). The Adult Religious Education and Faith Development project (AREFD) began in 2018 and is due to be completed in 2022. A brief overview of the origins, rationale, aims, and phases of the AREFD project will be provided as context. Secondly, as the paper focuses on Catholic education in the adult sector, the particularities of adult education as opposed to education in a school setting for younger people will be explored within the overall framework of lifelong learning. Thirdly, there will be a discourse upon the use of the term 'detraditionalized'. Thereafter, these three areas will be examined insofar as identifying recommendations emerging from the data provided by the participants in this specific phase of the AREFD project.

### 1.1. The Adult Religious Education and Faith Development (AREFD) Project

In October 2018, the Mater Dei Centre for Catholic Education (MDCCE), Dublin City University, launched the Adult Religious Education and Faith Development research project (AREFD). Funded by the Presentation Sisters North East Province, it was initially

a three-year project. However, the impact of COVID-19 resulted in an extension of the study until December 2022. The genesis of the AREFD project was twofold. Firstly, there was the awareness of existing work being carried out in communities across the island of Ireland, in both formal and informal contexts. Secondly, the project also has its roots in the understanding that there is a thirst and a hunger among adults to find new ways to deal with new questions as they emerge. These may have a spiritual or faith dimension, and may be connected with the various milestones and experiences that impact upon adults over the years.

Taking the first point further, it was clear that AREFD exists across Ireland and, in different ways, this work offers adults the opportunity to reflect on their faith, their religious understanding, their identity and their role in society. These initiatives, however, were not largely documented or collated in a cohesive manner. Anecdotally, there was a sense that the activities taking place for adults in terms of their religious education and faith development were carried out to some extent in isolation from each other. That is to say, that similar activities could be taking place in different areas but that those involved were not necessarily aware of each other and thus missing out on being able to share ideas, resources, and to reflect together perhaps on the efficacy of their work. There was also a sense that such opportunities were being provided in a broad manner, open to anyone interested, though often with the same people availing of them. This resulted in concerns amongst providers focusing on whether those actually most in need of adult religious education and faith development were in fact being reached. Additionally, there were queries as to how such work was contributing to long-term engagement of adults with faith and the depth of impact upon their lives. It was this 'culture of provision' that gave the research project its impetus. An early aim of the AREFD project was to in some way go further than anecdote, and through academic research and engagement with stakeholders, assess this culture of provision, investigate the overall opportunities available, and evaluate their efficacy in addressing the religious and spiritual needs of present-day Irish adults. This naturally dovetailed with the second issue driving the AREFD project, that of the interest in and desire for opportunities for adults to engage in religious education and faith development. An aim of the project was to facilitate a re-energising of adult religious education and faith development in Ireland by identifying the present needs of Irish adults and explore new and innovative avenues to address such needs. This would be a long-term goal, likely reaching beyond the timeframe of the AREFD project which would incorporate some pilot projects. However, it was hoped by the initiators of the project that the learning from it which would hopefully inspire work to continue 'on the ground' as it were through a somewhat ripple effect.

Over the lifetime of the project, the team are working according to a schedule of three phases. The first phase involved gathering empirical evidence on current practice in adult religious education and faith development in local faith communities in contemporary Ireland. During this time, the research team engaged in two central activities: commencing an extensive literature review and the construction of an online survey. The survey ran for six weeks in May/June 2019. It invited adults over eighteen across the island of Ireland to explore their understanding and experiences of religious education when they were at school but also bring them beyond that, to what matters to them in the present day. It encouraged them to reflect on how they express their beliefs and values; the opportunities (or lack thereof) for religious education/faith development at various stages of life; and, ultimately, what Irish adults would like to see happening in the future to engage them in ongoing religious education and faith development.

Moving into the second phase, the research team incorporated core insights from the ongoing literature review with the main themes emerging from responses to the online survey. Working in a qualitative framework for this phase, the research team sought to engage with a variety of groups and individuals to acquire a richer insight into their 'lived wisdom' gathered from their varied experiences in AREFD. The themes identified in the literature and from the survey provided a guide for an intentional conversation regarding

the broad and diverse spectrum of adulthood and possible opportunities for them to engage in religious education and faith development.

A total of fourteen semi-structured interviews/focus groups were conducted between December 2019 and April 2021 featuring twenty-two people from across the island of Ireland who have a wealth of experience in AREFD in diverse contexts. Those consulted looked back on the processes they have engaged in, the activities they have carried out, and the resources they have used. This was also a great opportunity for them to reflect on their achievements and challenges and consider how they might refine or adapt their work or even explore new aspects. Areas in which participants were engaged in adult religious education and/or faith development across both the Republic of Ireland and Northern Ireland included: retreat centres; pilgrimage; Catholic school management; academic research in religious education; training for voluntary pastoral ministry; evangelical ministry; diocesan advisors at primary and post-primary level; youth ministry; and parish ministry.

The third phase of the AREFD project focuses upon practical application of the learnings acquired in the form of supporting a small number of pilot projects. These pilot projects conducted with partnership communities and groups will serve as case studies, assisting in reviewing and strategically planning for best practice. Ultimately, the AREFD project aims to therefore provided evidenced-based proposals for new models and approaches responding to the adult education challenges of present-day Ireland with regard to religion and faith.

Selected findings the AREFD project up until the time of writing this article were featured in various conference presentations and a number of publications (Byrne and Sweetman 2021; Sweetman 2019a, 2019b, 2021a, 2021b). Information is also updated on the project website (https://www.dcu.ie/materdei-centre-catholic-education/adult-religious-education-and-faith-development-research-project, accessed on 12 October 2021).

### 1.2. The Specific Realm of AREFD

In looking towards reimagining adult religious education and faith development in a detraditionalised Ireland, it is important to be clear as to how the terms 'religious education' and 'faith development' pertain to the adult realm. This highlights the issues of education as formal, informal, and lifelong and the importance of the agency of the adult in his/her religious education and faith development.

The term 'religious education' in Ireland is usually associated with the formal educational activity in primary and post-primary schools. Looking specifically beyond the school years for examples of adult religious education, one might firstly recall the more formal examples such as courses of study, bible study groups, parish events, guest speakers, and so on. These usually have an explicit religious focus and belong within a particular denomination. It is these events that are generally seen to be provided, especially at certain times of the liturgical year such as Lent or Advent, and are open to whoever may be interested. Other formal education in religion for adults may be associated with training for ministry (Codd 2017), often taking place in seminaries or schools of theology (Stache 2014). These are not solely for ordination purposes, however, and terms such as lay theological education and lay ministry have come to the fore (Elias 2006). Elias, however, also emphasises that not all religious education takes place in formal settings, commenting that, 'the most pervasive adult religious education taking place today is informal education' (Elias 2012, p. 9). The literature shows a recognised trend of a growing number of adults seeking spiritual answers to their questions outside of formal religious institutions (Zeph 2000; Ó Murchú 1998). The breadth of adulthood, spanning decades if viewed chronologically and spanning depths if viewed in life experience, is vast. Consequentially, the formal education within primary and post-primary school is insufficient in itself as we encounter both good and bad times in our lives as we grow older, challenges as well as opportunities. Adults both need and deserve age-appropriate educational experiences throughout the course of their lives. In contrast to the more formal, perhaps traditional, provision of adult religious education, the rise in other, less formal opportunities is evident. In some cases,

these are not new occurrences but perhaps their status as opportunities for adults in this area, beyond the formal training for ministry, is somewhat more recognised. Informal religious education for adults can take place, for example, in homes, communities, places of worship, on pilgrimages or holidays, through group experiences, on the Internet and with mentors (Simojoki 2019; Chazan 2003; English 2000). In recent years, the online world is a particular context that has seen increasing research in relation to religious education, notably the contribution of Campbell (2004, 2005, 2010). It is recognised that not only are today's adults living in a plural and diverse culture when it comes to race, religion and creed but that their lives are 'digitally-infused' (Hutchings 2011). Online pedagogies open up fewer formal approaches to adult learning, allowing personalised, accessible, and flexible delivery according to participant needs and availability (Stuart-Buttle 2014).

'Religious education' and 'faith development' are distinct but interlinked. Particularly in relation to the breadth of adulthood, it is acknowledged that people are at different points in their lives and have different needs. Some may have grown up in a faith tradition, others not. Some may wish to explore their own faith tradition more, while others seek alternatives. Culture and society are also factors in how people may engage with their faith lives. The *General Directory for Catechesis* (Congregation for the Clergy 1997) was published to express the content of Christian faith within these changing contexts. The Irish Catholic Church responded, as other local Catholic Churches around the world did, in producing its own local directory, *Share the Good News* [SGN] (Irish Episcopal Conference 2010). The directory dedicates a full chapter to adult faith development and affirms its primary importance in facilitating members of the Church to grow into the fullness of lived Christian faith (SGN, paras. 68–90). Using SGN as its framework, the AREFD project employs the term faith development 'to encapsulate all the different approaches to ongoing education available to people from the beginning of their journey into Christian faith and throughout a lifetime of growth in that faith.' (SGN, para. 43).

The AREFD project adopts a relational approach to religious education and faith development, built on the importance of taking the person at whatever stage they are at, and encouraging real encounters with self, between educator/learner, and beyond. This emanates from an understanding of education as a lifelong endeavour:

> To be human means to learn. To be fully human entails a lifelong effort in acquiring knowledge, attitudes, skills and behaviours. The complexity of life and the constant changes that persons face, increasingly demand that adults continue to learn through their lives.
>
> (Elias 1993, p. 93)

Boschki (2005) states, 'religious education is a process that involves all dimensions of a person's relationships' (p. 115). Relationships change over time. All education, including religious education must therefore attend to all stages of a person's life, from birth to natural death. In Ireland and elsewhere, however, the focus of investment of resources, personnel and finance, has arguably been targeted toward young people. Responding to this issue, Byrne (2008) has pointed out that, 'Very often these young adults need assistance to move from the faith they learned as children to a personal appropriation of faith as adults.' (p. 38). Knowles (1973), called the adult learner 'a neglected species'. With so much emphasis placed on the transmission of the faith to and religious education of the young, it is easy to see his point. The challenge of adult religious education and faith development is the explicit acknowledgement of the adult as the active agent in his/her own learning. The culture of provision that has dominated the Irish landscape is an extension of the structured provision of such in the school setting. Adult religious education and faith development must begin from the position that the learner is an adult. Wickett (2005) advocates that theological education must 'consider the issues from the learner's perspective'. This places the adult in an active role where they must articulate their needs and their preferred modes of engagement. Remaining within a traditional culture of provision does not do the adult justice. Goodbourn (1996) shares this concern about not attending to the adult as learner, seeing the failure of the provider 'to take account of motivation and preferred method', the

'belief that adults don't want to learn' and the 'uncertainty as to how it is to be done' as barriers against adult Christian education.

### 1.3. Detraditionalisation

Boeve's (2012) explanation of detraditionalisation as a process in which a society's relationship with a prior set of norms changes may be applied to a variety of contexts. The particular contexts addressed in this paper are identity and agency.

There has long been a strong link in the minds of many between national identity in the Republic of Ireland population and the Christian religion, particularly Catholicism (Williams 2005; Fuller 2002). Culture and society in Ireland in recent years has embraced a greater diversity and plural reality as has been seen elsewhere (Kieran 2019; Anderson et al. 2016). In response, the state-sponsored curriculums at Junior Cycle (12–15-year-olds) and Senior Cycle (16–18-year-olds) were introduced in 2000 and 2003 (Department of Education and Science 2000, 2003), respectively. These were:

> Designed to encourage more than a comparative study of religions and beliefs, seeking to 'equip students to understand their own religious tradition or non-religious worldview and also to reflect on the religious traditions and worldviews of others'
>
> (Byrne et al. 2019, p. 204)

More recently, the publication in 2015 of the *Framework for Junior Cycle* (Department of Education and Skills 2015) demonstrates how the Irish education system is committed to responding best to the needs of its students, in this case Junior Cycle students, in an ever-changing, and detraditionalising society. In a landscape of shifting sands, the importance of religious education is even greater:

> Religious education provides a particular space for students to encounter and engage with the deepest and most fundamental questions relating to life, meaning and relationships. It encourages students to reflect, question, critique, interpret, imagine and find insight for their lives.
>
> (NCCA 2019, p. 6)

These assertions also pertain to the adult learner, however, the academic literature in the area of AREFD in the Irish context is only emergent.

Adults in present-day Ireland do not share the assumed identity-attribute of 'Catholic' or even 'religious' to the same extent as their predecessors in the mid- to late- twentieth century. The changing demographic in the Republic of Ireland can be seen in the census figures. The Central Statistics Office in one of its reports on the 2016 census confirms, '[ … ] that while Ireland remains a predominantly Catholic country the percentage of the population who identified as Catholic on the census has fallen sharply from 84.2 per cent in 2011 to 78.3 per cent in 2016'. (Central Statistics Office 2017, p. 72). Those of no religion now account for just under 10% of the population (9.8%), a considerable number compared with times past and even with 2011 (Central Statistics Office 2017, p. 72). There is a process of detraditionalisation in the relationship between the Irish adult and their religious, specifically Catholic, identity.

Secondly, the agency of the adult in brought into focus in relation to how active they may be in their own religious education and faith development. The clerical and hierarchical model of the church which has dominated for centuries disempowered many adults, gave them the sense that they have to meet some unspoken set of criteria in order to initiate some intentional AREFD in their lives (Byrne and Sweetman 2021). In addition, with a focus on adult religious education and faith development on formal training for ministry (Stache 2014; Codd 2017), the traditional culture of provision was overlooking the needs of both those who sought such engagement in a lay capacity as well as those whose sights were set on a more informal approach. If the adult is to be an active agent in his/her own religious education and faith development, it is necessary to have the required literacy to express their needs and the tools to manifest appropriate responses.

## 2. Methods

This paper reports on the second phase of the AREFD project. The aim of this phase was to consult with a diverse sample of individuals and groups involved in AREFD. By drawing on their wisdom and experience, the research team sought to identify key insights that would contribute, both at an academic and pastoral level, to the development of new AREFD opportunities. With a particular focus on reimagining adult religious education and faith development in a detraditionalised Ireland, the data from the participants was analysed with the aim of eliciting insights for such reimagining from their own lived experiences.

### 2.1. Procedure

The purpose of these interviews was to gather together the rich insights from the wealth of experience of the interviewees on the practicalities and possibilities central to adult religious education. The broad approach of an experiential mode of qualitative research (Braun and Clarke 2013) was therefore adopted with a list of open-end questions constructed by the research team. Open-ended questions in a semi-structured interview style was the chosen data collection methodology with the aim of encouraging the participants to lead the discussion. The questions served as prompts for the participants to indicate the nature of their involvement in AREFD either as a provider and/or participant. In particular, the strengths and weaknesses of any particular AREFD activities were to be elicited. Whilst gathering accounts of the breadth of experience in AREFD among the interviewees, the research team also adopted a critical approach to the research and invited the participants to reflect upon and critique their experiences with a view to making recommendations for the future development of AREFD. All participants were provided with plain language statements and informed consent forms in keeping with the protocol required by the DCU Research Ethics Committee. In-person interviews were audio-recorded. In the instance of interviews conducted via Zoom during COVID-19 restrictions, permission was sought to record the consultation according to the DCU Data Protection Unit protocols on conducting research online. All consultations took place between December 2019 and April 2021.

### 2.2. Participants

A purposive sampling strategy was employed to identify potential participants. Some participants had voluntarily contacted the research team in the earlier phase of the AREFD project expressing their interest to contribute in some way where possible. Other participants were approached where it was considered by the researchers that their context would complement the existing sample. The research team sought to engage with different forms of AREFD in a variety of contexts. In this case, context was understood as having three characteristics which assisted the research team in selecting the participants. Firstly, it referred to size of the projects in which the different people were involved. Secondly, their geographical location (i.e., local/nationwide, rural/urban, etc.) was an element. Thirdly, the overall mode or genre of the AREFD taking place was identified. Cognisant of the timeframe of the project and the capacity of the research team, a smaller number of participants from as diverse a range of contexts was preferable to a large sample. Twenty-two people working in adult religious education and faith development, in both the Republic of Ireland and Northern Ireland and across a variety of settings were consulted in total in this second phase of the AREFD project. This comprised of ten individual interviews and four focus groups. Twelve males and ten females took part. Seven participants were Catholic clergy or in a Catholic religious order with the remaining fifteen identifying as lay Catholics. Contexts in which they have worked included retreat centres, pilgrimage sites, Catholic school management, academic research in religious education, training for voluntary pastoral ministry, evangelical ministry, diocesan advisory services at primary and post-primary level, youth ministry, and parish ministry.

*2.3. Analysis*

All interviews were transcribed by the post-doctoral researcher. Transcripts were anonymised using labels for participants (e.g., P1, P2) and focus groups (e.g., G1, G2). Following Braun and Clarke (2013), the research team read the transcripts and familiarised themselves with the data. Data extracts were identified that encapsulated the sentiments and opinions that were stated either most frequently or with the greatest amount of conviction, assertion and importance to the participant. This phase of the analysis was data-driven and open coded. Manual thematic analysis was then conducted over a number of reviews to recode for broad themes. Applying knowledge of the literature, the research team interpreted these broad themes and further reviewed the data to form distinct themes for reporting. Data extracts were chosen to illustrate these themes in the dissemination stage. For this paper, four themes were chosen for discussion.

## 3. Results

The participants were generous with their insights, sharing their stories of how they became involved in adult religious education and faith development and what may have been the various highpoints and learning curves of this process. Drawing from various contexts, in roles as both providers and participants, the learnings from these interviewees are hereby applied in four themes. The four themes are offered as principles by which a reimagining of adult religious education and faith development in Ireland could take place.

*3.1. Focus on Jesus Christ*

> We have to use the name of Jesus as much as we can, but too often, like he's not referred to at all. If you listen to Bishops giving interviews and all of that, they talk about the church. Jesus is not mentioned.

> (P15)

A cursory interpretation of detraditionalisation may be at odds with this theme. Boeve (2012), however, clarifies that in the process of detraditionalisation, the relationship is changed, not eroded. Based on the viewpoints of the participants, as extracts here attest, the relationship that adults have with Jesus should be rediscovered, not replaced. In other words, it should be reimagined. The participants recounted how for young children in school, Jesus is the dominant feature in religious education. Jesus is presented as a real person, with real encounters. They recalled their involvement in sacramental preparation where Jesus was central. The interviewees noted how in their work with children, the children would speak about Jesus as if they knew him personally, just as they know their neighbours on the street. However, they also pointed out that they felt the real person of Jesus became less central in religious education as time progressed. More emphasis emerged on moral issues such as abortion, divorce, and women priests, for example.

> Because to use the business language, you know, your USP to Jesus is all we have. There's no other unique selling point for us because other places create community. Other places care for people who are vulnerable. Other places, do all of these other things and some of them are an awful lot more competent than we are. So, what is unique to us? The one thing that we don't mention. And then we wonder.

> (P15)

As P15 notes, Jesus should be central in adult religious education and faith development. There is an argument whereby given the prevalence of Jesus in the curriculums for younger children, a movement towards debates on moral issues and so on might seem more mature and age-appropriate. The Christian, however, clearly cannot leave Jesus behind. Instead, the relationship, so strongly featured in the faith development of young people, especially in the sacraments of initiation, must be reimagined as an adult relationship. One participant highlighted the absence of this relationship, or at least the awareness of it, in their perception of the people they worked with:

> But the idea of having a personal relationship with Jesus Christ wasn't something that they really had an understanding of.
>
> (P16)

This focus on Jesus emerged strongly in the interviews, suggesting that in these changing times, a tangible relationship with Jesus is important:

> Something that people that touches people. Something really speaks to them in their life, and that is Christ centered.
>
> (P9)

### 3.2. Physicality of Religion

Related to this concept of a tangible relationship with Jesus, participants highlighted the importance of seeing faith as physical. This has two dimensions, how one is invited into engagement with religious education and faith development, and how this engagement then impacts upon one's actions and attitudes.

> Our religion is physical. It's actually a physical religion that is engaged with the senses. It's not ... if we keep it in our heads, it becomes a political discussion and you opt in and you opt out ... If we try to impress with our knowledge, we will have no impact whatsoever.
>
> (P15)

> Gets them out of their head ... it's not just about mindset. It's also about heart sense.
>
> (P1)

Knowledge is one aspect of religious education and faith development. While participants acknowledged the importance of learning about beliefs, traditions and reading Scripture, the deeper connection was emphasised. 'Senses' and 'heart' were both mentioned a number of times as a means to clarify that the participants felt it was important to acknowledge how religious education and faith development is more than facts, tenets and text. This may be related to the strong association amongst adults with religious education as a school-based activity. Placing a strong and intentional emphasis on the senses and an embodied approach to religious education and faith development, however, was seen as significant by the participants.

> Every part of their body is engaged, mental, emotion, physical ...
>
> (P2)

> And until we see our God as physical ... it's very hard then to reject and not to engage.
>
> (P15)

Religious education and faith development also has a physical impact according to the participants. Through their engagement with AREFD, they are changed, and consequently, their actions and behaviours are affected.

> If we say we believe this and we these are beliefs and these are our values, how do we how do we make that tangible? So it's through our behaviours and actions.
>
> (P4)

> it's something that has really inspired me, informed me, motivated me.
>
> (P9)

### 3.3. Investment and Intentionality

Participants indicated the prevalence of providing resources, such as talks and course materials, which is a central feature in the culture of provision. They, however, raised the issue of purposefully investing in the people, rather than the product. Instead of continuing

to create resources and provide materials, the interviewees highlighted how there needs to be investment of time and finance in the people involved in AREFD, either as providers or participants.

> How much are we investing in the building and how much of investing in the people? And it's about 99 percent. One percent. Yeah. It's beginning to shift in some places, but it's a whole new culture, which is a challenge. How we break down that culture.
>
> (P15)

> The mistake we make is really preparing excellent resources, but not resourcing I think, the most important thing the person who will be at the center of delivering.
>
> (P10)

This investment of time and finance is to be seen as more than a form of training. There is an element of placing trust and encouragement in the people who come forward with ideas for and an interest in AREFD. The participants pointed to the importance of recognising the gifts and skills of adults, nurturing these skills and where necessary investing either in terms of time and/or finance.

> In fact, it's equipping people with skills that are relevant to our time and necessary for our time ... often we go in because we do everything for people robbing them of their own capacity and their own skills, kind of going in as the experts.
>
> (P6)

There was a sense amongst the participants that the adult realm of religious education and faith development can sometimes be slightly left to chance, perhaps as a hopeful offshoot in particular for parents who present their children for the sacraments of initiation.

> You're just trying to nurture that so they can nurture their children. We have a very strong belief here that it's through the children that we will reach the adults.
>
> (P11)

Intentionally facilitating adults as adults in their own right rather than in role as parent, for example, emerged as significant in the interviews. In doing so, it was seen as important by the interviewees that proper investment should be made in terms of payment, job status and ongoing professional development.

> So, they have to prioritize and figure out and be creative around wherever those finances come from. And what would the job descriptions look like for those people
>
> (P12)

> the biggest thing that can happen in any team is if the relationships aren't right and there isn't an understanding that people think different, or work different or have different passions for things so that you learn how to facilitate, particularly, you know, when you're working between the clergy and the laity and you're trying to empower the laity
>
> (P11)

### 3.4. Simplicity and Clarity

The interviewees noted that working with adults can be challenging since the expansive spectrum of adulthood with all its milestones, experiences and possibilities can at times seem overwhelming. In comparison to school-based religious education which has a structure, there is a greater freedom with adult religious education and faith development. A consequence of this freedom, however, is that there it can be difficult to focus on particular needs and accept that it is not possible to cover everything or reach everyone. Instead, they emphasised the importance of keeping things simple—doing less but doing it well.

we can complicate our motivation. You know, what do we want? We want people to be more engaged. What does 'engaged' mean?

(P15)

if you keep something simple enough and keep it wholesome, pure and what they're going to get out of it, not a big song and dance. Keep it straight and simple. I think that's what works here. I think that's what works here. And people see that and they see that it's not twaddle, it's not—it's real. It works, and they want to be involved in it.

(P13)

I think that when a vision is clear and when there is joy, then there will be life and there will be fruit.

(P15)

## 4. Discussion and Recommendations

As Boeve (2012) noted, detraditionalisation is a process. Therefore, it is ongoing and dynamic. In acknowledging that AREFD is undergoing a process of detraditionalisation, as it is an aspect of Irish society and culture that is evolving as a whole, it is therefore important to purposefully reflect upon the purpose and forms of AREFD. The participants carried out such reflections as part of the consultative phase of the AREFD project. It was evident, however, and in fact named by some, that such reflective practice was not necessarily commonplace even to them. Though involved in different ways in AREFD over the years, few of the participants indicated that they rarely paused to assess and evaluate their work. Instead, they moved from one programme to the next, lamenting the sometimes poor uptake on their offerings, but nonetheless not purposefully reflecting and strategizing. This only served to perpetuate the culture of provision. Overall, the data from the consultative phase, as reported upon in this paper, suggest the need for a culture-shift in AREFD. This nourishment of a culture-shift is the reimagining of AREFD in a detraditionalised Ireland.

The first reimagining of AREFD is in terms of clarifying the identity of the adult. Within the traditional culture of provision, the dominant offering of AREFD was towards the adult in a particular role. More often than not, this role was in some way parental, with a goal of equipping the adult to pass on the faith. The data suggests however that is necessary to go deeper than such roles and open up AREFD to the adult at whatever stage of faith and life he/she is. In tandem with this is the acknowledgement that more needs to be carried out to develop the skills of religious literacy for adults (see Parker 2020) If they are being granted greater agency to engage in AREFD, at whatever stage they are at and then applying their learnings to whatever role(s) they may face, then they must be given the space and the tools to explore, reflect upon, and articulate their spiritual and religious needs. Many adults are stuck in the religious language of their youth from formal engagement in school-based RE and/or catechesis. Adult religious literacy is therefore a prerequisite with any reimagining of AREFD in a detraditionalised context.

The second reimagining of AREFD relates to affirming its value in its own right. The traditional culture of provision of AREFD whereby courses are run or talks offered, implies that uptake is not strategically planned for. Engagement depends upon the availability of the prospective participant, scheduling amidst a busy lifestyle, juggling other commitments and financial obligations. Evidently, more intentionality and investment are given to formal school-based religious education and sacramental preparation. It is of course important when working with adults to be invitational and welcoming. However, if AREFD is to be truly valued in its own right, then sincere consideration must be given to how its status is portrayed. The most common attempts at engaging adults can be at the times of their children's preparation for certain sacraments. Leaving the addressing of the adults spiritual and religious needs to chance in some ways devalues those very needs. Instead, recognising the hunger and desire that exists amongst adults for religious education

and faith development, strategic planning is called for. Investment in suitably qualified personnel who specific task focuses upon adult religious education and faith development is recommended. The continued support for those personnel and for suitable resources is also encouraged in an overall framework of lifelong learning. It would be beneficial to have further conversation specifically articulating adults' understanding of 'religious education' and 'faith development' in order to focus efforts and offer opportunities appropriately.

The final, and most central, reimagining of AREFD in a detraditionalised Ireland is the rediscovery of the Person of Jesus Christ. Unifying the key themes that emerged from the data, the Person of Jesus Christ is the simple and clear focus of a physical and personal relationship open to all adults. Participants spoke about the strength of the relationship that children have with Jesus, especially at times of sacramental preparation. On the one hand, some respondents indicated that this relationship can weaken, with some adults focusing more on issues, particularly moral concerns, as opposed to maintaining a personal relationship with Jesus. On the other hand, some indicated that they witnessed the most powerful transformations in people engaging with AREFD when those people rediscovered, or reimagined, their personal relationship with Jesus. The physicality of religion as a theme speaks clearly to an appreciation for emotional, mental, and physical engagement; to mind, body and spirit and to head, heart and hands. In relationships, these senses are amplified and developed. Affirming the understanding of education as contributing to the full development of the human being, it is recommended therefore that the Person of Jesus Christ be central to any reimagining of AREFD in a detraditionalised Ireland.

## 5. Conclusions

The adults of the present were the children of the past, and the children of present are the adults of the future. It would be a disservice to the person if the bulk of his/own religious education and faith development opportunities were concentrated upon in the relatively short time-span that relates to formal school-based settings. The AREFD project confirms the importance and value of opportunities for religious education and faith development for adults at whatever stage of their faith and lives they are. The project also highlights the need for reflective practice and investment of time, personnel and finances for AREFD. The data reported upon in this paper offers some insights by those who have worked most closely in the area in recent years as to how we may work towards a reimagining of AREFD in a detraditionalised Ireland.

**Funding:** The AREFD project conducted at the Mater Dei Centre for Catholic Education, Dublin City University is funded through a gift agreement between the Presentation Sisters North East Province and the DCU Educational Trust. This study constitutes part of the overall AREFD project and did not receive any external funding.

**Institutional Review Board Statement:** Not applicable.

**Informed Consent Statement:** Informed consent was obtained from all subjects involved in the study.

**Data Availability Statement:** Interview transcripts are not publicly available.

**Conflicts of Interest:** The author declares no conflict of interest.

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
