# Peer review of "Reimagining Adult Religious Education and Faith Development in a Detraditionalised Ireland"

_religions, doi:10.3390/rel12110963_

Round 1

Reviewer 1 Report

Unify the length of „spaces“ between sentences (between the comma ending one sentence and the first capital letter of the next one) - in other cases there is shorter space (gap).

  1. 161-176- theory (one paragraph) on „faith development“ is very short and unbalanced comparing with religious education – actually there is little about various phases of faith development – at least in various stages of adulthood. We suggest it would be good to at least list the stages. The word „faith develpment“ is a part of the title of the article that is why we are pointing at this (at least to mention faith development in adulthood generally, e.g. Stages of faith development by Fowler or Sparkman
  2. 245 – „the agency of the adult in brought into focus“? Something missing in the sentence...

l.300 – why some were interviewed individually and some in focus groups? (focus groups have a different dynamic than individual interviews, did they not influenced the results?)

  1. 409: n fact, it's it's (repeated in this form?)

In the part of discussion – very few resources discussed with, e.g. there are studeies on religious literacy – it would be good to refer to some of them (e.g. try S. Parker - Religious literacy: spaces of teaching and learning about religion and belief, 2002)

Reviewer 2 Report

I very much appreciated this article. My only suggestion is about formatting. When quoting the interviews, It was difficult to identify where a quote ended and a new paragraph began. Using bullet points (*) to indicate quotes would rectify this.
